# Understanding the diversity of the metal-organic framework ecosystem

Seyed Mohamad Moosavi [1,2], Aditya Nandy [2,3], Kevin Maik Jablonka [1], Daniele Ongari [1], Jon Paul Janet[2], Peter G. Boyd [1], Yongjin Lee [4], Berend Smit [1✉] & Heather J. Kulik [2✉]

Millions of distinct metal-organic frameworks (MOFs) can be made by combining metal nodes and organic linkers. At present, over 90,000 MOFs have been synthesized and over 500,000 predicted. This raises the question whether a new experimental or predicted structure adds new information. For MOF chemists, the chemical design space is a combination of pore geometry, metal nodes, organic linkers, and functional groups, but at present we do not have a formalism to quantify optimal coverage of chemical design space. In this work, we develop a machine learning method to quantify similarities of MOFs to analyse their chemical diversity. This diversity analysis identifies biases in the databases, and we show that such bias can lead to incorrect conclusions. The developed formalism in this study provides a simple and practical guideline to see whether new structures will have the potential for new insights, or constitute a relatively small variation of existing structures.

[1] Laboratory of Molecular Simulation, Institut des Sciences et Ingénierie Chimiques, École, Polytechnique Fédérale de Lausanne (EPFL), Rue de l'Industrie 17, Sion CH-1951 Valais, Switzerland. [2] Department of Chemical Engineering, Massachusetts Institute of Technology, Cambridge, MA 02139, USA. [3] Department of Chemistry, Massachusetts Institute of Technology, Cambridge, MA 02139, USA. [4] School of Physical Science and Technology, ShanghaiTech University, 201210 Shanghai, China. ✉email: berend.smit@epfl.ch; hjkulik@mit.edu

The fact that we have an exponentially increasing[1–4] number of different MOFs ready to be tested for an increasing range of applications opens many avenues for research[5,6]. However, this rapid increase of data presents concerns over the chemical diversity of these materials. For example, one would like to avoid screening a large number of chemically similar structures. Yet, the way the number of materials evolves is prone to a lack of diversity[7,8]. For example, one can envision an extremely successful experimental group focusing on the systematic synthesis of a particular class of MOFs for a specific application. Such successes may stimulate other groups to synthesise similar MOFs, which may bias research efforts towards this class of MOFs. In libraries of hypothetical MOFs, biases can be introduced by algorithms that favour the generation of a specific subsets of MOFs. At present, we do not have a theoretical framework to evaluate chemical diversity of MOFs. Such a framework is essential to identify possible biases, quantify the diversity of these libraries, and develop optimal screening strategies.

In this work, we introduce a systematic approach to quantify the chemical diversity of the different MOF libraries, and use these insights to remove these biases from the different libraries. The focus of our work is on MOFs, as for these materials there has been an exponential growth of the number of studied materials. However, the question on how to correctly sample material design space holds for many classes of materials.

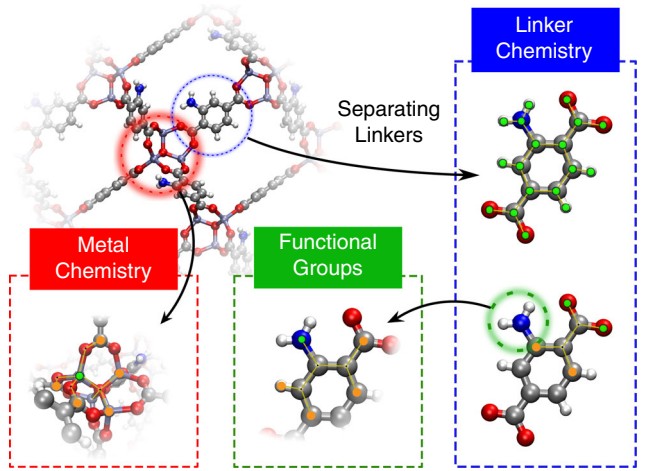

**Fig. 1 Description of the three domains of MOF chemistry.** Metal centre RACs are computed on the crystal graph. Linker and functional-group RACs are computed on the corresponding linker molecular graph. Linker chemistry includes two types of RACs, namely full linker and linker connecting atoms. The graphs show the start atom (in green) and the nearby atom (in orange) used to define the RACs descriptors (see the "Methods" section).

## Results

**Development of descriptors for MOF chemistry.** One of the aims of this work is to express the diversity of a MOF database in terms of features that can be related to the chemistry that is used in synthesizing MOFs as well as generating the libraries of hypothetical structures. At present, different strategies have been developed to represent MOFs with feature vectors[9–12]. However, the global material descriptors[9,13–16] that are presently used are not ideal for our purpose. We would like to directly connect to the structural building blocks of MOFs, which closely resemble the chemical intuition of MOF chemists, in which a MOF is a combination of the pore geometry and chemistry (i.e., metal nodes, ligands and functional groups)[6,17]. However, it is important to note that in developing these descriptors, it is impossible to completely separate the different effects and scopes. For example, for some MOFs adding a functional group can completely change the pore shape. Hence, depending on the details of the different types of descriptors and properties of interest, this may be seen as mainly pore-shape effect, while other sets of descriptions will assign it as functional-group effect.

To describe the pore geometry of nanoporous materials we use simple geometric descriptors, such as the pore size and volume[18]. For the MOF chemistry, we adapt the revised autocorrelations (RACs) descriptors[19], which have been successfully applied[19–22] for building structure–property relationships in transition metal chemistry[19,23]. RACs are discrete correlations between heuristic atomic properties (e.g., the Pauling electronegativity, nuclear charge, etc.) of atoms on a graph. We compute RACs using the molecular or crystal graphs derived from the adjacency matrix computed for the primitive cell of the crystal structure (see the "Methods" section). To describe the MOF chemistry, we extended conventional RACs to include descriptors for all domains of a MOF material, namely metal chemistry, linker chemistry, and functional groups (Fig. 1 and the "Methods" section).

**Description of the databases.** We consider several MOF databases: one experimental and five with in silico predicted structures (see Supplementary Note 2 for more details of databases).

The Computation-Ready, Experimental (CoRE)[2,24–26] MOF database represents a selection of synthesised MOFs.

The first in silico generated MOF database (hMOF) was developed by Wilmer et al.[3] using a "Tinkertoy" algorithm by snapping MOF building blocks to form 130,000 MOF structures. This Tinkertoy algorithm, however, gave only a few underlying nets[27]. An alternative approach, using topology-based algorithms has been applied by Gomez-Gualdron et al.[28] for their ToBaCCo database (~13,000 structures), and by Boyd and Woo[4,29] for their BW-DB (over 300,000 structures). A comprehensive review of this topic can be found here[30].

We use CoRE-2019 and a diverse subset of 20,000 structures from the BW-DB (called BW-20K) to establish the validity of the material descriptors. In addition, a relatively small database of around 400 structures developed by Anderson et al.[14] (ARABG-DB) was included for comparison with their conclusions about importance of structural domains[14]. For this test, we focus on adsorption properties as their accurate prediction requires a meaningful descriptor for both the chemistry and pore geometry. We study the adsorption properties of methane and carbon dioxide. Because of their differences in chemistry (i.e. molecule shape and size, and non-zero quadrupole moment of carbon dioxide), designing porous materials with desired adsorption properties requires different strategies for each gas. To emphasize on these differences, we study the adsorption properties at three different conditions, namely infinite dilution (i.e. Henry regime), low pressure and high pressure.

**Predicting adsorption properties of MOFs.** We first establish that our descriptors capture the chemical similarity of MOF structures. As a test we show that instance-based machine-learning models (kernel ridge regression (KRR)) using these descriptors can accurately predict adsorption properties. A KRR model with a radial basis function kernel uses only similarity that is quantified using pairwise distances in the feature space; hence, the performance of the model can demonstrate the validity of the descriptors. KRR models show good performance in predictions of the adsorption properties of CoRE-2019 and BW-20K databases (see Supplementary Note 3 for parities and statistics). We

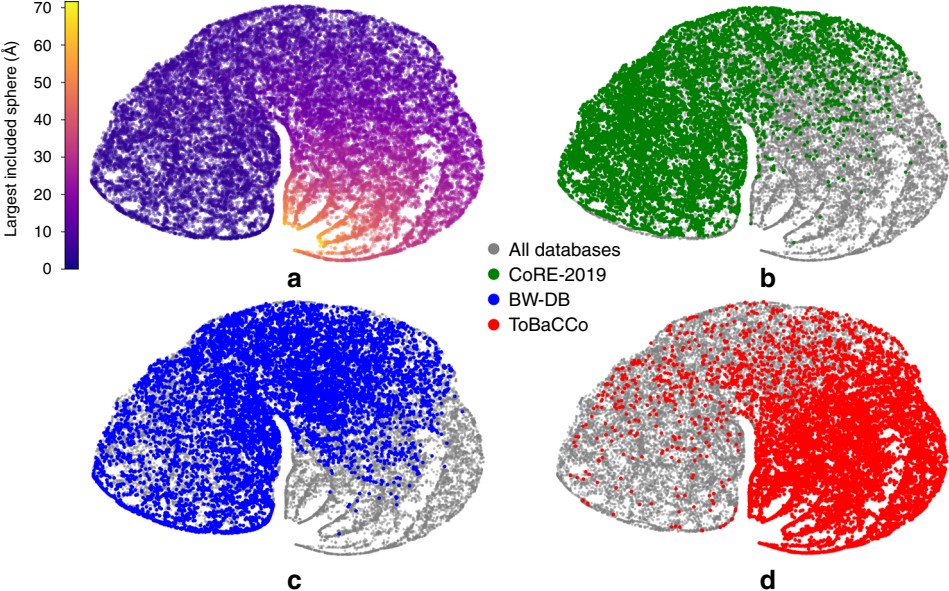

**Fig. 2 Map of the pore geometry of MOFs.** To project the geometric descriptor space of MOFs to a 2D map we use the t-distributed stochastic neighbour embedding (t-SNE)[67] method (see Supplementary Note 6 for principal component analysis (PCA)). The t-SNE method preserves local similarity, ensuring similar structures are mapped close to each other in two dimensions. **a** The current design space colour coded with the largest included sphere. In (**b**), (**c**), and (**d**), the green, blue and red dots are representing the materials in the CoRE-2019, BW-DB and ToBaCCo databases, respectively, which are overlaid on the design space represented in grey. PCA plots show a similar distribution of databases (see Supplementary Note 6).

observe that for those properties that are less dependent on the chemistry, e.g., the high-pressure applications of $CH_4$ and $CO_2$, the geometric descriptors are sufficient to describe the materials with the average relative error (RMAE) in the prediction of the gas uptake being below 5%. In addition, if we compare the relative ranking of the materials, we also obtain satisfactory agreement as expressed by the Spearman rank correlation coefficient (SRCC) above 0.9. On the other hand, for the applications where chemistry plays a role, e.g., the Henry coefficient of $CO_2$, the chemical descriptors are essential to accurately predict the materials properties (RMAE ~ 5% and SRCC ~ 0.8). The performance and accuracy of our models is comparable with the prior studies[14,31–35] (see a comprehensive list in ref. [36]). However, to be able to compare the accuracy and performance of different models and feature sets, one needs to perform a benchmark study using a fixed set of materials with high diversity and their corresponding properties as for example, we observe the performance of machine-learning models varies considerably from one database to another.

The significance of the chemical descriptors is further illustrated by the predictions of the maximum positive charge (MPC) and the minimum negative charge (MNC) of MOF structures (SRCC above 0.9 and 0.7, respectively). The geometric descriptors are nearly irrelevant for these charges (SRCCs below 0.5 for all cases). This explains the relatively poor performance in prediction of $CO_2$ adsorption properties at low pressures using only geometric descriptors as electrostatic interaction plays a crucial role. This analysis shows that our RACs and geometric descriptors are meaningful representations for the chemical space of MOFs for both $CH_4$ and $CO_2$ adsorption over the complete range of pressures. As a consequence, if two materials have similar descriptors, their adsorption properties will be similar. Hence, we can now quantify how the different regions of design space are covered by the different databases.

**Diversity of MOF databases**. We define the current chemical design space as the combination of all the synthesized materials

and the in silico predicted structures, i.e., all the materials in the known databases. The real chemical design space, of course, can be much larger, as one can expect that novel classes of MOFs will be discovered. It is instructive to visualize how each MOF database is covering the current design space. This design space, as described by our descriptors, is a high-dimensional space and to visualize this we make a projection on two dimensions.

The projection of the pore geometry of our current design space is shown in Fig. 2a. The colour distribution shows a gradient in the pore size of the MOFs, from small to large pores moving on the map from left to right. Other panels show how the different MOF databases are covering this space. The distributions of the geometric properties of the databases are considerably different from each other (Fig. 2b–d). For example, the experimental MOFs (CoRE-2019) are mainly in the small pore region of the map. Remarkably, the hypothetical databases also have very different distributions. While BW-DB covers the intermediate pore size regions, ToBaCCo is biased to the large pore regions of the design space.

The hypothetical structures have been generated to explore the design space of MOFs beyond the experimentally known structures. In Fig. 3 we show how these databases are covering the design space (see Supplementary Note 6 for the distribution of each database and for PCA method). We use diversity metrics[37] to quantify the coverage of these databases in terms of variety (V), balance (B) and disparity (D). The pore geometry, linker chemistry and functional groups design spaces are well covered and sampled by the hypothetical databases. However, we observe a serious limitation in diversity, in particular in the variety of the metal chemistry in hypothetical databases (Fig. 3b). Compared with the experimental database, the variety of the metal chemistry of MOFs by hypothetical databases is surprisingly low; only a limited number of MOF metal centres are present (18 metal SBUs for all hypothetical databases, see Supplementary Note 14).

The choice of the organic linker and the placement of functional groups are readily enumerated; one can take the large databases of organic molecules[38] as a rich source of the possible

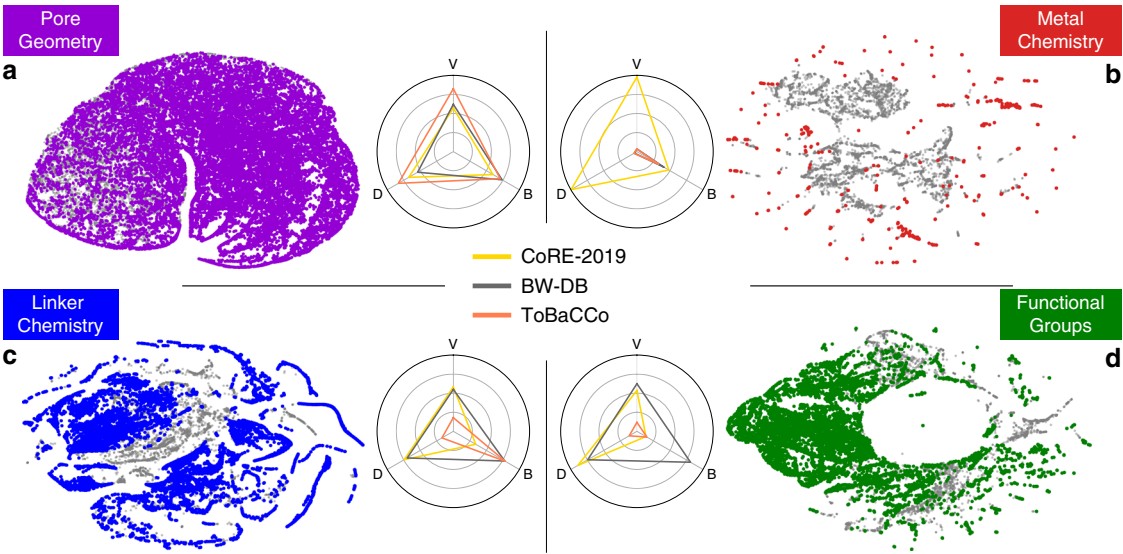

**Fig. 3 Diversity metrics and maps of different domains of MOF structures.** The t-SNE method was used to project the **a** pore geometry, **b** metal chemistry, **c** linker chemistry and **d** functional groups descriptor spaces to 2D maps. Only descriptors up to the second coordination shell were included for metal chemistry to emphasize the local metal chemistry environment. In each panel, the structures from the hypothetical databases are coloured and overlaid on the entire known design space represented in grey. The radar charts show the three diversity metrics: variety (V), balance (B) and disparity (D), for the three databases. For this analysis, first we discretize the space into a fixed number of bins. Variety measures the number of bins that are sampled, balance the evenness of the distribution of materials among the sampled bins, and disparity the spread of the sampled bins (see the "Methods" section for more details).

MOF linkers or functional groups. In contrast, the metal nodes of MOFs are typically only known after a MOF is synthesised. For example, at present we cannot predict that if Zinc atoms during the MOF formation would cluster in a Zinc paddle-wheel (e.g., in Zn-HKUST-1)[39], a single node (e.g., in ZIFs)[40], $Zn_4O$ (e.g., in IRMOFs)[6], or to a totally new configuration.

The diversity in metal chemistry was further reduced by the choice of researchers and/or limitations in the MOF structure assembly algorithms. For example, some of the hypothetical MOF databases are deliberately focused on specific sub-classes of MOFs to systematically investigate structure–property relationships. For example, the study by Gomez-Gualdron et al.[41] that focuses on generating stable MOFs using Zirconium-based metal nodes for gas storage, Witman et al.[42] on 1-D rod MOFs featuring open-metal sites for $CO_2$ capture, and Moosavi et al.[43] on ZIFs with various functional groups and underlying nets for the mechanical stability. Lastly, in silico assembly of MOFs possessing complex nodes that are connected via multiple linkers, especially on a low-symmetry net, is still challenging for the current structure generation algorithms[44]. Therefore, we expect that there are many missing points on the metal chemistry map in Fig. 3b which will be found in the coming years.

**Applications of diversity analysis.** We illustrate the importance of quantifying the diversity of the different databases by three examples. The first example illustrates how machine learning can be used to extract insight on how the performance of a material is related to its underlying structure[14,19,21]. As our descriptors represent each domain of the MOF architecture, we can quantify the relative importance of these domains on $CH_4$ and $CO_2$ adsorption.

Within each database, the importance of variables varies significantly across different gases and different adsorption conditions (see Supplementary Note 5). These results follow our intuition; the chemistry of the material is more important in the low-pressure regime, while at high pressures the pore

geometry is the dominant factor. Moreover, we observe that material chemistry is more important for $CO_2$ than for $CH_4$ adsorption.

If each of these databases would have covered a representative subset of MOF chemistry, one would expect that each database would give a similar result for the importance of the different variables. However, we observe striking differences when we compare across different databases. An illustrative example is $CO_2$ adsorption at low pressure. Anderson et al.[14] concluded from their analysis of the (ARABG-DB) database that the metal chemistry is not an important variable for $CO_2$ adsorption. However, Fig. 4a shows that for each of these databases different material characteristics are important for the models in predicting $CO_2$ adsorption. For example, pore geometry is the most important variable in the BW-20K, while metal chemistry in CoRE-2019, and the functional groups in ARABG-DB. Since the material properties were computed using a consistent methodology for all databases, these differences in the importance of variables originate in the differences in the underlying distribution of material databases (see Fig. 3 and Supplementary Note 6 for distribution of databases). For instance, the reason why metal chemistry was not identified as an important factor by Anderson et al. was that metal chemistry was not explored sufficiently in their database as only four SBUs were used for structure enumeration. Also, since these values are the relative importance, one can argue that in CoRE-2019 MOFs, the functional groups were not exploited as much as metal chemistry. At this point, it is important to note that our analysis is based on the current state-of-the-art methods that is used in screening studies, i.e., generic force fields and rigid crystals. It would be interesting to see how improvements in, for example, the description of open-metal sites in MOFs will change this analysis. If the changes are large, such improvements will likely have a large impact.

In our second example, we focus on how our diversity analysis can help us to identify opportunities for the design of new

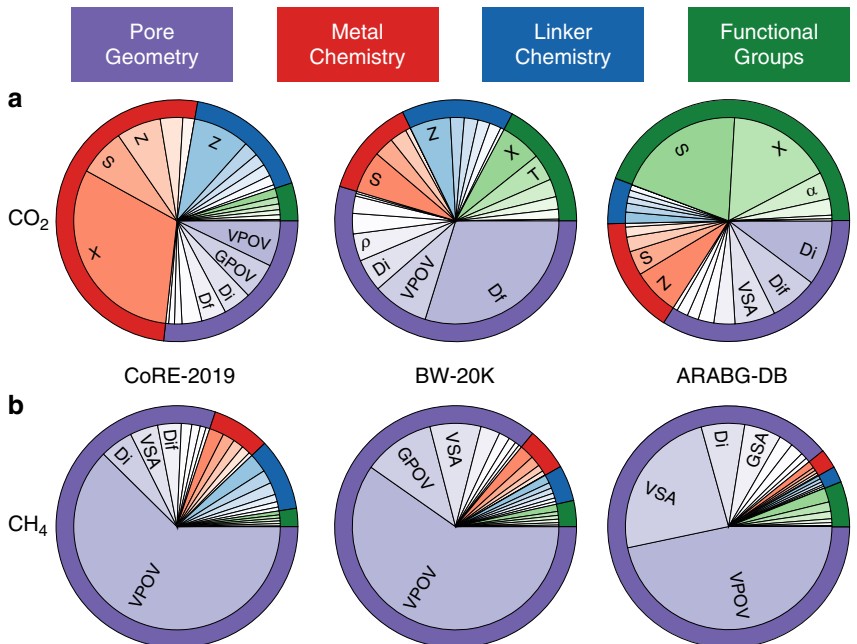

**Fig. 4 Database dependence of the importance of material characteristics.** Pie charts showing the SHapley Additive exPlanations (SHAP) values (importance of variables) for **a** the low-pressure $CO_2$ adsorption and **b** $CH_4$ deliverable capacity. SHAP values were computed for the random forest regression models using a training set of CoRE-2019 and BW-20K, and all structures in ARABG-DB. For the chemical features, the importance of variables was summed over all RAC depths for each of the heuristic atomic properties. See the "Methods" section for the meaning of the labels. Similar values for importance of variables were obtained using other techniques (see Supplementary Note 5).

structures. At present, there are over 90,000 MOFs that have been synthesised and one would like to be sure that MOF 90,001 adds relevant information. Similarly, for the hypothetical databases one would add new structures to any screening study only if they are complementary to the many that already exist.

For $CO_2$ capture from flue gases, which corresponds to $CO_2$ adsorption at low pressure in our study, we have shown that metal chemistry cannot be ignored (Fig. 4a). Our diversity analysis shows that this domain is not well covered by hypothetical databases (see Fig. 3). Therefore, exploring different metal chemistries in these databases would increase the diversity of these databases. For this we have developed a methodology to mine unique MOF building blocks from the experimental MOF databases (see the "Methods" section). In Supplementary Note 7, we show some of these SBUs that have not been used for structure enumeration in these hypothetical databases yet, and including these missing structures in a screening study could lead to the discovery of materials with superior performance.

For methane storage our analysis shows that the single most important factor is the pore geometry (see Fig. 4b). All databases confirm that pore geometry is the most important factor. For this application, each of the databases have a sufficient diversity in geometric structures and other factors do not matter. This observation provides an important rationale for the provocative conclusion of Simon et al.[45] that there is no point in looking for new structures for methane storage as they are not expected to perform significantly better for this application. Simon et al. arrived at this conclusion from a large screening of 650,000 random selection of structures from many databases of different classes of nanoporous materials. Our study shows that indeed a large selection of structures from different databases will cover the entire geometric space of the current design space. To significantly outperform the best performing materials one would need a completely new chemistry and mechanism, e.g., framework flexibility[46].

In the final example, we focus on the effect of bias in the databases on the generalisability and transferability of machine-learning predictions. Intuitively, one would expect that if we include structures from all regions of the design space in our training set, our machine-learning results should be transferable to any database. We illustrate this point for the two databases CoRE-2019 and BW-DB. We randomly select 2000 structures that we use as test set. A diverse set of structures based on the chemical and geometric descriptors was obtained from the remaining structures in these two databases[47,48]. The accuracy of random forest models trained using this diverse set is compared with the models trained using training sets from each database in Fig. 5. Clearly, the models that were trained on databases which are biased to some regions of the design space result in poor transferability for predictions in unseen regions of the space. In contrast and not surprisingly, the model that is trained with a diverse set performs relatively well for both databases. Besides, the diversity in training set lead to a more efficient learning. In supplementary materials, we show the learning curves that demonstrate the models trained on the diverse set have systematically lower error than the ones trained using biased databases. The number of training points in which the learning curves plateau can be an indication of the minimum number of structures for optimal coverage of the design space for a particular application. This number is obviously proportional to the complexity of the material property, i.e., how many materials characteristics are affecting the materials properties.

## Discussion

An interesting side effect of MOF chemistry is that the enormous number of materials makes this field ideal for big-data science. This development raises all kinds of new, interesting scientific questions. For example, we have now so many experimental and hypothetical materials that brute-force simulations and experiments are only feasible on a subset of materials. Hence, it is

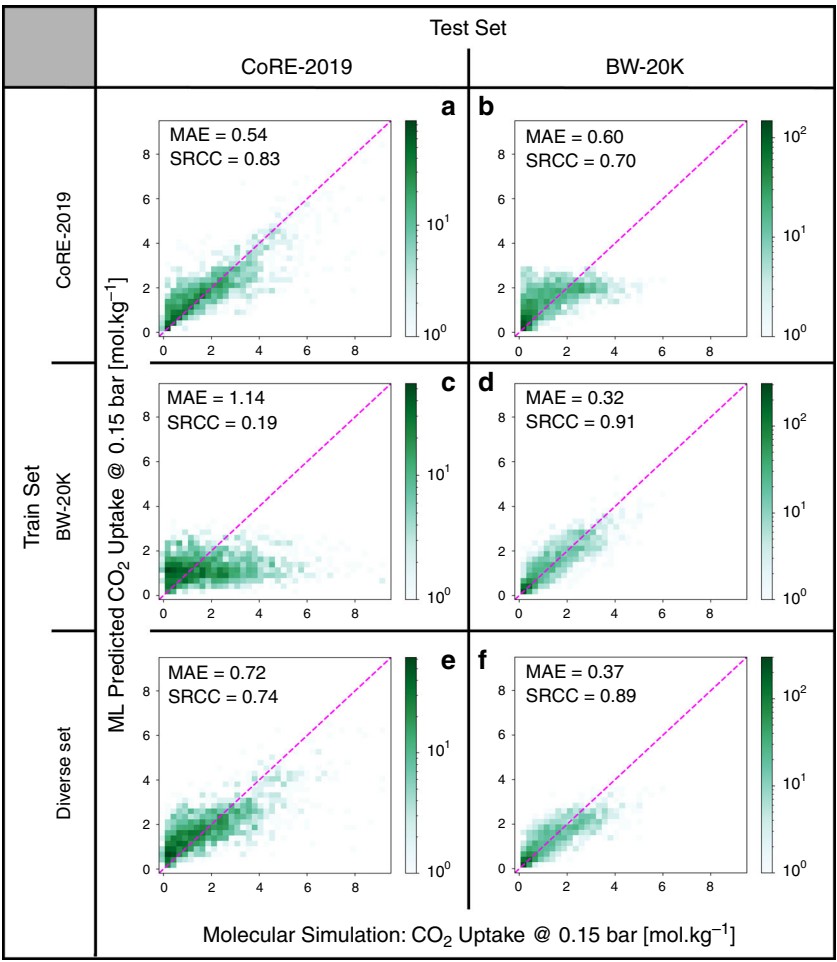

**Fig. 5 Impact of diversity in training data on transferability of models.** The parity plots of random forest models using full features; rows and columns correspond to the training and test sets, respectively. The dashed lines represent the parity. The size of training sets is equal in all cases. The same structures were used as test sets in each column. The diverse set was selected using the MaxMin[47] algorithm using all geometric and chemical descriptors. The colour bars show the number of structures in each cell of the histograms.

essential that this subset covers the relevant chemistry as optimally as possible. In this work, we have developed a theoretical framework on how to arrive at the most diverse set of materials representing the state of the art of MOF chemistry.

Our framework relies on the notion that for chemists the chemical design space of MOFs is a combination of pore geometry, metal nodes, organic linkers and functional groups. By projecting a material on a set of relevant descriptors characterizing these four domains of MOF chemistry, we can quantify the diversity of databases. Adding structures that increase the diversity metrics, implies that these structures add new information to the database. Given that there are already so many materials and databases, there is a need for a simple and powerful practical guideline to see whether new set of structures will have the potential for new insights or are relatively small variations of existing structures. Analysis of the diversity can also give us insights in parts of the chemical design space that are not fully explored. An interesting historical perspective is shown in Fig. 6, in which we plot as metric of novelty of the discovered materials the distance to the geometry descriptor of the previously discovered materials. Here, we assume pore geometry is the important factor of interest. The jumps in the graph nicely identifies structures that opened a new direction of MOF research[5,49–53], where 2012 was an exceptionally good year, which

include the discovery of the IRMOF-74[53] series and the material with the lowest density[51] and highest surface area[49] at their time.

One cannot separate diversity from the application. For example, if one is interested in the optical properties of MOFs, which largely depends on charge transfer between metal and ligand species, diversity in pore geometry might not be that important, and for such a screening study the optimal representative set of materials will be different from say, a gas adsorption study. Yet, the same procedures to generate such a diverse set can be used provided that the properties depend sufficiently gradual on the relevant descriptors. If one has a property that dramatically changes by a slight change of the structure of the MOF, our method would flag these structures as similar while the properties are in fact very different. Of course, once such property is identified one can re-weight the measure of similarity to ensure that those aspects of the descriptors that can distinguish these materials carry more weight.

In this work we aim to address the question whether a new material adds novelty. We try to develop transparent and objective criteria to quantify how different a novel material is with respect to the state of the art. However, as soon as we use this for a particular application, it becomes subjective. For example, if Fig. 6, we selected novelty of pore geometry. This measure by definition completely ignores, for example, the importance of making

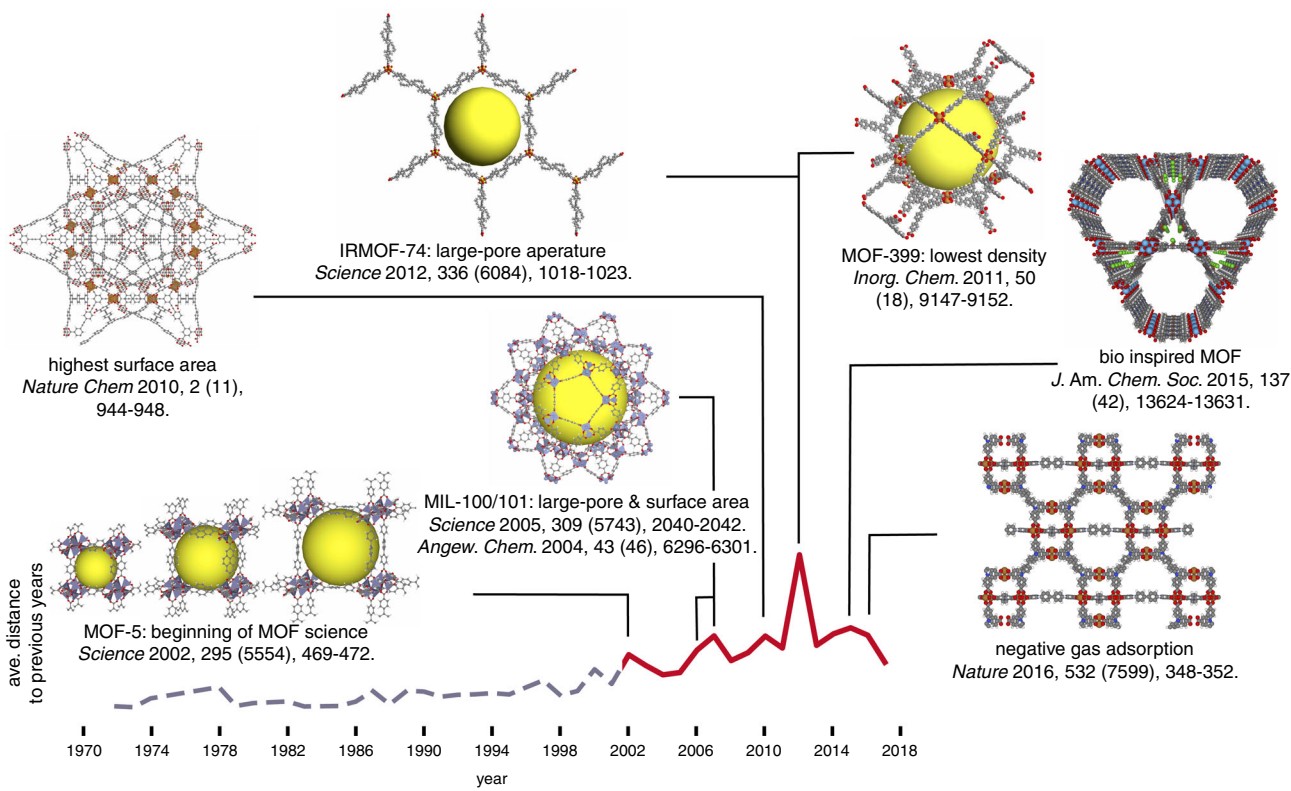

**Fig. 6 Timeline of evolution of MOF geometry.** For each year, the average of relative distance in the geometry descriptor space to the MOFs reported in Cambridge structural database (CSD)[26] in the preceding years is shown with red line. The MOFs with largest distance for some of the peaks are shown in the inset[5,49–53,68,69]. The years on the timeline are corresponding to the year that a structure has been deposited in CSD. The grey line shows the coordination polymers reported in CSD before the beginning of the MOF chemistry as a separate field of research, shown in red.

the first MOF with a particular metal, which might be the single most important factor for, say, an application related to catalysis.

MOF chemistry is not a static field; new classes of MOFs will be constantly developed. The protocol that was introduced in this work can be (trivially) extended in the future to include these new MOFs as they get reported, allowing to always generate a set of most diverse structures that is representative of the whole database of known structures.

## Methods

**RACs for MOFs.** RACs[19] are products and differences on the graph of heuristic atomic properties. RACs were first introduced for machine-learning open shell transition metal complex properties[19,20,23]. The relative importance of heuristic properties proved valuable for interpreting structure–property relationships and similarity of these transition metal complexes[21]. We have devised an approach to extend RACs to periodic MOF materials by dividing MOFs into their constituent parts. A typical[19] difference-based RAC correlation is computed on the graph representation of the structure using:

$$\underset{scope}{\overset{start}{P}}{}^{diff}_d = \sum_i^{start} \sum_j^{scope} (P_i - P_j)\delta(d_{i,j}, d). \tag{1}$$

In this equation, atomic property $P$ of atom $i$ selected from *start* atom list is correlated to atom $j$ selected from *scope* atom list when they are separated by $d$ number of bonds. To devise MOF chemistry-specific RACs, we extend the concepts of *start* and *scope* introduced[19] for metal-centred and ligand-centred RACs in transition metal complexes. Two atom lists, namely *start* and *scope*, are needed to compute these RACs (Eq. (1)). For the metal-centred RACs, we use the crystal graph as the *scope* atom list and the *start* atom list only includes all metals (see Supplementary Note 1 for full list). These RACs thus emphasize the metal and SBU contributions to MOF chemistry and property prediction. In describing linkers and functional groups, we use RACs computed on the molecular graph of the corresponding linker. In this approach, we only correlate atoms on the same linker, and therefore, the *scope* atom list includes all the atoms from the same linker of the

starting atoms. To construct the molecular graph for each linker, we start by splitting the MOF to the corresponding linker lists. Removing the metals from the crystal graph gives us a set of floating connected components. We remove the atoms that are only bonded to the metals and/or hydrogens, e.g., the bridging oxygen in $Zn_4O$, and the corresponding hydrogen that are connected to these atoms, leaving us with only the organic linkers and the coordinated organic molecules to the metal centres. By separating the subgraphs of these connected components, we obtain the molecular graph for each linker. Linker chemistry is described with two *start* atom lists, including full linker and linker connecting atoms. Full linker atom list includes all the atoms on the linker. Linker connecting atoms are the atoms that have a chemical bond with a metal centre. Lastly, any atom on a linker that is not a carbon or hydrogen atom, and is not linker connecting atom is assigned to be a functional group and is included in the *start* atom list for functional-group descriptors. Note that carbon-based functionalisations, e.g., methyl functionalisation, would not be identified as a functional group in this approach.

Similar to applications of RACs on transition metal complexes[19–21], five heuristic atomic properties, including atom identity ($I$), connectivity ($T$), Pauling electronegativity ($\chi$), covalent radii ($S$) and nuclear charge ($Z$) were used to compute RACs. To this set, we add polarisability ($\alpha$) of atoms for the linker descriptors as suggested[14] to be an important factor for gas adsorption properties of MOFs. These properties are used to generate metal-centred, linker and functional-group descriptors. Lastly, we take the averages of these descriptors to make a fixed length descriptor. In total, this analysis produces 156 features (see Supplementary Note 1 for details).

Lastly, we apply our unique graph identification algorithm (see below) on the linkers and store the simplified molecular-input line-entry system (SMILES) string (converted using OpenBabel)[54,55] of the unique linkers for further featurisation and exploratory data analysis of MOF databases. Moreover, we flagged structures that might have chemical inconsistency in the linker chemistry using RDKit[56].

**Mining building blocks.** The approach explained in the previous section can correctly identifies the organic SBUs. However, rigorously recognising inorganic SBUs is challenging, requires advanced methods, and might be dependent on the crystal graph simplification method[17]. In this study, we leverage a method to mine inorganic SBUs specific to our data set. We make an atom list including metal centres and their first and second coordination shells. We extract inorganic SBUs

by separating all connected subgraphs after removing all the atoms which are not included in this list from crystal graph. Finally, we identify unique organic and inorganic SBUs by removing all isomorph labelled molecular graphs using Cordella et al.'s[57] approach as implemented in NetworkX[58].

**Molecular simulation.** The adsorption properties of the materials were computed assuming rigid frameworks. The guest–guest interactions and host–guest interactions were modelled using Lennard–Jones potential truncated and shifted at 12.8 Å and Coulombic electrostatic interactions computed by Ewald summation. The force-field parameters of the framework atoms and gas molecules were extracted from UFF and TraPPE force fields, respectively (see full list of parameters in Supplementary Note 11 and 12), using the Lorentz–Berthelot mixing rule for pairs. Partial atomic charges of framework atoms were generated using EQeq[59]. Grand canonical Monte Carlo and Widom insertion were used to compute gas uptake and Henry coefficient of the materials, respectively. Each calculation consists of 4000 initialisation cycles followed by 6000 equilibrium cycles. All the gas adsorption calculations were performed in RASPA[60]. Adsorption properties were computed at 0.15 bar (5.8 bar) and 16 bar (65 bar) for $CO_2$ ($CH_4$) for low and high pressures, respectively. All adsorption calculations were performed at room temperature. The pore geometry was described using eight geometric descriptors, namely largest included sphere ($D_i$), largest free sphere ($D_f$), largest included sphere along free path ($D_{if}$), crystal density $\rho$, volumetric and gravimetric surface area and pore volume. The geometric descriptors were computed using Zeo++[18,61], using a probe radius of 1.86 Å.

**Machine learning.** Random forest regression (RF), gradient boosting regression (GBR) and kernel ridge regression (KRR) models were used in this study. All computations were performed in scikit-learn[62] machine-learning toolbox in python.

The hyperparameters for GBR and RF models were chosen by grid search optimisation using 10-fold cross-validation (CV) minimising the mean absolute error (see Supplementary Note 8 and 9 for the range of hyperparameters). For the KRR models, we first perform feature selection. Both recursive feature addition (RFA) and explained variance threshold methods were used to find the the feature subset that minimises the 10-fold CV mean absolute error of the model. For the RFA method, the order of feature addition was done based on the importance of features derived from the random forest mean decrease in impurity importance of variables following the strategy in ref. [23]. The hyperparameters of the KRR models were chosen by minimising the 10-fold CV score of the model using a mixed optimisation methods, including Tree of Parzen Estimators (TPE), annealing and random search, using the hyperopt[63] package.

The features and labels were centred to zero and scaled using their mean and standard deviation, respectively. Train-test splitting was performed randomly and the size of the train sets are mentioned in the caption of each parity plot or table in the main text and the Supplementary Notes. All the statistics reported were computed by averaging over 10 different random seeds used for train-test splitting except in the figures for transferability of models between databases where fixed test sets were used.

The relative importance of variables was computed for the random forest models. Three different approaches were used to derive the feature importance (see Supplementary Note 5 for comparison). The first approach is based on the mean decrease in impurity (Gini importance) which is computed while training a random forest regression. The second and third approach are permutation importance and SHapley Additive exPlanations (SHAP)[64], respectively, which were computed for the test or train set.

**Diversity metrics.** To compute the diversity metrics, we first split the high-dimensional spaces into a fixed number of bins by assigning all the structures to their closest centroid found from k-means clustering. Here, we use the percentage of all the bins sampled by a database as the variety metric. Furthermore, we use Pielou's evenness[65] to measure the balance of a database, i.e., how even the structures are distributed among the sampled bins. Other metrics, including relative entropy and Kullback–Leibler divergence are a transformation of Pielou's evenness and provide the same information (see Supplementary Note 16 for comparison). Here, we use 1000 bins for these analyses (see sensitivity analysis to the number of bins in Supplementary Note 16). Lastly, we compute disparity, a measure of spread of the sampled bins, based on the area of the concave hull of the first two principal components of the structures in a database normalized with the area of the concave hull of the current design space. The areas were computed using Shapely[66] with circumference to area ratio cutoff of 1.

## Data availability
Supplementary Information is available for this paper. The analysed structures with the partial charges, features and labels for machine learning, SMILES strings of MOF linkers, feature importance analysis, exploratory data analysis plots, diversity metrics, timeline, and the force-field parameters that were used and are needed to reproduce this study are deposited on the Materials Cloud archive via https://doi.org/10.24435/materialscloud:3y-gr. Correspondence and requests for additional materials should be addressed to berend. smit@epfl.ch and hjkulik@mit.edu.

## Code availability
The code for parsing, featurization and identifying unique building blocks of MOF structures is available free of charge on molSimplify program Github (https://github.com/hjkgrp/molSimplify). All codes are available under GNU General Public License v3. The script for selecting a diverse subset of materials using MaxMin method is available on the Materials Cloud archive via https://doi.org/10.24435/materialscloud:3y-gr.

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

## Acknowledgements

This study was supported by the Swiss National Science Foundation (SNSF) with a Doc. Mobility fellowship to S.M.M. (grant number P1ELP2_184404). S.M.M., K.J., D.O. and B. S. are supported by the European Research Council (ERC) Advanced Grant (grant agreement no. 666983, MaGic) and the National Center of Competence in Research (NCCR), Materials' Revolution: Computational Design and Discovery of Novel Materials (MARVEL). H.J.K., A.N. and J.P.J. are supported by a Defense Advanced Research Projects Agency Young Faculty Award (grant D18AP00039). This material is based upon work supported by the National Science Foundation Graduate Research Fellowship under Grant No. 1122374 (to A.N.). The authors would like to thank Diego Gomez-Gualdron for providing support in interpreting the ToBaCCo database used in this work.

## Author contributions

S.M.M. and A.N. developed the RACs featurisation code. K.M.J. performed time-evolution analysis. S.M.M., A.N., K.M.J. and J.P.J. developed the machine-learning workflows. S.M.M., D.O., Y.L. and P.G.B. computed the adsorption properties and analysed the databases. S.M.M., B.S. and H.J.K. designed the project. All authors contributed to the analysis of the data. S.M.M., A.N., K.M.J., B.S. and H.J.K. wrote the paper with the contribution from all authors.

## Competing interests

The authors declare no competing interests.
