## [Peer Review File · Nature Communications]

REVIEWERS' COMMENTS:

Reviewer #1 (Remarks to the Author):

The authors quantify the diversity of chemical space in common MOF databases by using machine learning techniques. While this is a very interesting topic - which is furthermore relevant for a wide community -, I am a bit disappointed in the actual results and conclusions of the study. The analysis is definitely a welcome and necessary addition to the proliferation of MOF databases, but it does not provide substantially new insights as suggested in the abstract and introduction. Many of the observations were already known from earlier studies, but are now combined into an exhaustive study. However, I appreciate how they highlight some of the issues in MOF screening studies with a number of well-chosen examples, and it is a nice wake-up call for all computational scientists who routinely use one specific (MOF) database.

In conclusion, it is an instructive manuscript that clearly points out some potential problems of screening studies, and it is supported by a lot of data and explanatory figures. Nevertheless, as the current manuscript is written very specifically for MOFs, I am not convinced by the general importance of this work for the more general audience Nature Communications aims at and I feel that this manuscript might be better suited in a more specific materials science/physical chemistry journal. I leave it to the Editor to decide based on this journal aims and scope.

Below a number of suggestions are given that could help the authors to strengthen their work:

1. The authors have chosen to study CH₄ and CO₂ adsorption because of their differences in chemistry. Furthermore, they decide to work in three different adsorption regimes (infinite dilution, low pressure, high pressure). It is well-known that generic force fields are typically unable to describe the chemically specific interactions between the framework and CO₂. For example, this is the case for MOFs with coordinatively unsaturated metal sites. Selecting a different generic force field could deliver very different results. Achieving quantitative and qualitative correct predictions for the right reasons is therefore a challenging task. Of course, this is the price that has to be paid when working with large databases. However, how can the authors be sure that their analysis will not change when selecting a different force field? How sensitive is e.g. the analysis on the importance of the different variables (Figure 4) on the chosen level of theory?
2. Many observations mentioned in the manuscript (especially in the section on 'Predicting adsorption properties of MOFs') are already known in the MOF community. The authors should refer to previous literature.
3. The authors select a diverse set of structures from the CoRe-2019 and BW-DB database in their third example to illustrate that models trained on such a subset possess a good transferability. I think that generating diverse subsets without bias towards some regions is a very interesting product of their study, and I believe such subsets should be used in the future as sanity checks for screening studies. Is it possible to provide a tool to automatically sample such a set of structures from a given larger database?
4. One of the central questions of this study is about how one would like to be sure that a new MOF adds relevant information. Can the authors provide an easy (quantitative or qualitative) way for experimental researchers to test a new structure to the large set of structures that they considered?

Reviewer #2 (Remarks to the Author):

In this excellent contribution the authors develop tools to quantify and analyze the structural and chemical diversity MOF databases and apply these tools to some prominent MOF databases providing an evidence-based perspective on past large-scale simulation studies on MOFs, which will be probably eye-opening for many readers. The authors take advantage of the available information about MOF diversity to answer questions regarding how well the MOF-space has been explored in previous high throughput screening efforts, to what extent bias is present in different databases, what the implications of these biases are on

conclusions that emerged from earlier studies, and what are key ways to improve diversity and reduce bias in future studies. The manuscript is filled with numerous useful “bits” of information that I am sure will guide future molecular simulation and machine learning studies in MOFs, in a way that will probably make MOF discovery more effective. In my opinion, this is the kind of work that I would expect to find in *Nature Communications*, so I am recommending the publication of this manuscript. I have, however, some comments that I would like the authors to consider. They are presented below not necessarily in order of importance.

1) I am just curious about why no neural network models were considered for the machine learning part of this work. Was there a particular rationale for the chosen machine learning methods? Was it because it is less straightforward to examine variable importance?

2) I think that the higher metal diversity of the CoRE MOFs, or otherwise the lack of metal diversity in hypothetical databases is something that has been pointed out here and there in the past. In that regard, the mining of nodes for future generation of MOFs is really informative. I wonder if the authors have any thoughts on why the metal diversity is so low in hypothetical databases. Sometimes I am under the impression that CoRE MOFs have numerous metals, but in single atom form with “flimsy” linker connections, such as only one oxygen of a carboxylate, that lead to easy collapse upon activation. This might be a reason why so many CoRE MOFs remain in obscurity, while most of the experimental MOF literature overwhelmingly focus on the usual suspects (MOF-5, MOF-74, UiO-66, Cu-BTC, MIL-100, ZIF-8, etc). In this case, it would seem to me that hypothetical databases tend to emphasize on “robust” nodes from well-known MOFs that are known to withstand activation. Other times, though, I am under the impression that the connectivity of CoRE MOFs is simply much more complicated than what MOF generation methods can handle. Can the authors comment?

3) How common are the nodes presented in the SI? That is how many CoRE MOFs have them? What kind of linkers are associated with them?

3) About the discussion around Figure 4. Specifically, the ARABG database. In the cited publication, Anderson et al report variable importance for some CO₂ adsorption metrics (selectivity, working capacity), with a pore descriptor having the largest importance (kind of in line with the variable importance here for BW-20K). These are not the CO₂@0.15bar adsorption metrics the authors are using here, but it does make me wonder how sensitive general conclusions regarding whether chemistry or pore geometry are more important are to how the chemical and geometrical descriptors are defined, as well as the type of machine learning method.

4) One implication of the discussion in this work is that simply trying to get high number of structures in a database for the sake of getting a high number is not the best use of resources. A more targeted MOF generation for the application of choice may be the smarter strategy. For example, the ToBaCCo database seems to have been originally explored for H₂, CH₄ and Xe/Kr, which are not expected to be affected by functionalization, making it OK for the database not to have functional groups. On the other hand, the ARABG database seems small but emphasizes on functional groups, which base on the discussion of this paper seems very important for CO₂ adsorption. For ARABG, machine learning was done with a small set (considering typical databases), so my question is whether in the current work the authors have unveiled some sort of, at least semi-quantitative, insights or rules of thumb about critical database size (assuming maximum diversity) beyond which machine learning models will not learn much more. (This perhaps relates to the kind of work presented in Fig. 5)

4) I think it would be a good idea to present histograms in the supplementary information as complementary way to present the information given in Figure 2 and Figure 3. Some readers may find those histograms easier to interpret.

5) Arguably, one way to read Fig. 4 could be that the screening the CoRE MOFs fails to capture that linker functionalization is important for CO₂ adsorption as noted by screening of ARABG or BW-20K, or as thought of my experimentalists who, for example, consider strategies such as amino functionalization to boost CO₂

sorption. In inspecting Fig S13 and S14, I can't help but notice that the CoRE MOFs fail to cover an area of functional groups that is covered by the BW database (I don't know about ARABG). Moreover, the area the CoRE MOFs do cover tend toward the area that ToBACCo covers, which we know does not really have functional groups. So I wonder if the CoRE MOFs are just lacking in those key functional groups that truly have an impact on CO₂ adsorption, which allow metal chemistry to take such an outsized role.

6) The authors state: "This explains the relatively poor performance in prediction of CO₂ adsorption properties using only geometric descriptors as electrostatic interaction plays a crucial role." Based on an earlier sentence, I think the authors may have meant to say: "(...) prediction of CO₂ adsorption properties at low pressure".

Reviewer #3 (Remarks to the Author):

The authors have created a well written paper that describes a new way to classify and organize the many MOFs available in popular databases, and to study their diversity in doing so. In addition to being well written, the paper is well organized and the results are clearly presented. I think this paper can be accepted essentially without revisions.

That being said, notions of MOF diversity are necessarily (?) subjective. It's hard to imagine how one can be write or wrong when measuring MOF diversity, so the aim is more to be more or less useful, but that will then depend on what applications MOF scientists are interested in. To their credit, the authors do get into applications and comment on the utility of their measures for those purposes... but it would have been nice to see an at least short discussion in the introduction that acknowledges this fundamental subjectivity. Conversely, if I am wrong and MOF diversity can be objectively measured, then I would certainly want the authors to explain that early in the paper!

In summary, unsurprisingly good work from great authors. Looking forward to seeing it published!

Response to Reviewer #1

The authors quantify the diversity of chemical space in common MOF databases by using machine learning techniques. While this is a very interesting topic - which is furthermore relevant for a wide community -, I am a bit disappointed in the actual results and conclusions of the study. The analysis is definitely a welcome and necessary addition to the proliferation of MOF databases, but it does not provide substantially new insights as suggested in the abstract and introduction. Many of the observations were already known from earlier studies, but are now combined into an exhaustive study. However, I appreciate how they highlight some of the issues in MOF screening studies with a number of well-chosen examples, and it is a nice wake-up call for all computational scientists who routinely use one specific (MOF) database.

In conclusion, it is an instructive manuscript that clearly points out some potential problems of screening studies, and it is supported by a lot of data and explanatory figures. Nevertheless, as the current manuscript is written very specifically for MOFs, I am not convinced by the general importance of this work for the more general audience Nature Communications aims at and I feel that this manuscript might be better suited in a more specific materials science/physical chemistry journal. I leave it to the Editor to decide based on this journal aims and scope.

Below a number of suggestions are given that could help the authors to strengthen their work:

1. The authors have chosen to study CH₄ and CO₂ adsorption because of their differences in chemistry. Furthermore, they decide to work in three different adsorption regimes (infinite dilution, low pressure, high pressure). It is well-known that generic force fields are typically unable to describe the chemically specific interactions between the framework and CO₂. For example, this is the case for MOFs with coordinatively unsaturated metal sites. Selecting a different generic force field could deliver very different results. Achieving quantitative and qualitative correct predictions for the right reasons is therefore a challenging task. Of course, this is the price that has to be paid when working with large databases. However, how can the authors be sure that their analysis will not change when selecting a different force field? How sensitive is e.g. the analysis on the importance of the different variables (Figure 4) on the chosen level of theory?

Authors reply:

The reviewer is absolutely right that the details of the outcome of our analysis for a particular application will depend on the quality of the simulated (or experimental) isotherms. If there are systematic errors in the force field, our methodology will correctly reproduce these incorrect results. In the revised manuscript we have added a sentence to emphasize this point. In this context we also emphasize that our analysis is a snapshot of our current knowledge and that the idea is to constantly update our analysis. Important in this is that we can then exactly assess the impact of, say, an improvement of the force field describing open metal sites.

“At this point it is important to note that our analysis is based on the current state of the art methods that is used in screening studies, i.e., generic force fields and rigid crystals. It would be interesting to see how improvements in, for example, the description of open metal sites in MOFs will change this analysis. If the changes are large, such improvements will likely have a large impact.”

2. Many observations mentioned in the manuscript (especially in the section on ‘Predicting adsorption properties of MOFs’) are already known in the MOF community. The authors should refer to previous literature.

Authors reply:

In the revised manuscript, we included more citation to prior works in this section.

3. The authors select a diverse set of structures from the CoRe-2019 and BW-DB database in their third example to illustrate that models trained on such a subset possess a good transferability. I think that generating diverse subsets without bias towards some regions is a very interesting product of their study, and I believe such subsets should be used in the future as sanity checks for screening studies. Is it possible to provide a tool to automatically sample such a set of structures from a given larger database?

Authors reply:

With the revised manuscript, we provide the following tools and features that can be used in the future studies:

1. The code for finding the diverse set is included to the materials cloud repository. The code takes as input the feature set and the selected feature columns and provide the diverse subsample using MaxMin method.
2. The feature set provided on materials cloud now contains the SMILES strings of the linkers and the chemical composition of the MOFs. The SMILES strings can be useful for further featurising the linkers and also exploratory analysis of the MOF databases for various applications.
3. We flagged structures that might be chemically invalid. We use RDKit to check the valence of atoms on organic linkers for sanity check. These are typically the structures in CoRE-MOF database that have missing hydrogen, error in occupancy, etc.

Changes in the method section:

“Lastly, we apply our unique graph identification algorithm (see below) on the linkers and store the simplified molecular-input line-entry system (SMILES) string (converted using open babel) of the unique linkers for further featurisation and exploratory data analysis of MOF databases. Moreover, we flagged structures that might have chemical inconsistency in the linker chemistry using RDKit.”

, and in the code availability section:

“The script for selecting a diverse subset of materials using MaxMin method is available on the Materials Cloud archive via: <https://doi.org/10.24435/materialscloud:3y-gr>”

4. One of the central questions of this study is about how one would like to be sure that a new MOF adds relevant information. Can the authors provide an easy (quantitative or qualitative) way for experimental researchers to test a new structure to the large set of structures that they considered?

Authors reply:

Yes, we agree that this would be desirable, and we are working on developing a simple web app for this. However, it is not yet in a stage we can make it public. All codes and data that we used in this study are provided in the materials cloud so researcher can use it, but we agree this is not easy for experimental scientist yet.

Response to Reviewer #2

In this excellent contribution the authors develop tools to quantify and analyze the structural and chemical diversity MOF databases and apply these tools to some prominent MOF databases providing an evidence-based perspective on past large-scale simulation studies on MOFs, which will be probably eye-opening for many readers. The authors take advantage of the available information about MOF diversity to answer questions regarding how well the MOF-space has been explored in previous high throughput screening efforts, to what extent bias is present in different databases, what the implications of these biases are on conclusions that emerged from earlier studies, and what are key ways to improve diversity and reduce bias in future studies. The manuscript is filled with numerous useful “bits” of information that I am sure will guide future molecular simulation and machine learning studies in MOFs, in a way that will probably make MOF discovery more effective. In my opinion, this is the kind of work that I would expect to find in Nature Communications, so I am recommending the publication of this manuscript. I have, however, some comments that I would like the authors to consider. They are presented below not necessarily in order of importance.

1) I am just curious about why no neural network models were considered for the machine learning part of this work. Was there a particular rationale for the chosen machine learning methods? Was it because it is less straightforward to examine variable importance?

Authors reply:

A more flexible model, e.g., neural network (NN), would perhaps reach a higher accuracy in the supervised tasks, here in prediction of the adsorption properties. However, our focus in this work was on unsupervised tasks, i.e. the diversity analysis. A key element for our diversity analysis was to establish that the descriptors, as they are, are meaningful representation of the materials characteristics such that the pairwise distances in the descriptor space are meaningful. An instance-based machine learning model, e.g. kernel ridge regression (KRR), fulfills this criterion as it uses only pairwise distances in feature space for predictions. Conversely, A NN model transforms this representation in a non-linear way that the original distances lose their meaning. To make it clearer, in the revised manuscript, we include the underlined text:

“We first establish that our descriptors capture the chemical similarity of MOF structures. As a test we show that instance-based machine learning models (kernel ridge regression (KRR)) using these descriptors can accurately predict adsorption properties. Such instance-based model uses only similarity that is quantified using pairwise distances in the feature space; hence, the performance of the model can demonstrate the validity of the descriptors. KRR models show good performance...”

2) I think that the higher metal diversity of the CoRE MOFs, or otherwise the lack of metal diversity in hypothetical databases is something that has been pointed out here and there in the past. In that regard, the mining of nodes for future generation of MOFs is really informative. I wonder if the authors have any thoughts on why the metal diversity is so low in hypothetical databases. Sometimes I am under the impression that CoRE MOFs have numerous metals, but in single atom form with “flimsy” linker connections, such as only one oxygen of a carboxylate, that lead to easy collapse upon activation. This might be a reason why so many CoRE MOFs remain in obscurity, while most of the experimental MOF literature overwhelmingly focus on the usual suspects (MOF-5, MOF-74, UiO-66, Cu-BTC, MIL-100, ZIF-8, etc). In this case, it would seem to me that hypothetical databases tend to emphasize on “robust” nodes

from well-known MOFs that are known to withstand activation. Other times, though, I am under the impression that the connectivity of CoRE MOFs is simply much more complicated than what MOF generation methods can handle. Can the authors comment?

Authors reply:

We agree with the reviewer that several factors have led to too little metal diversity in hypothetical databases. In the revised manuscript, we added these factors (see underlined text):

“The choice of the organic linker and the placement of functional groups are readily enumerated; one can take the large databases of organic molecules as a rich source of the possible MOF linkers or functional groups. In contrast, the metal nodes of MOFs are typically only known after a MOF is synthesised. For example, at present we cannot predict that if Zinc atoms during the MOF formation would cluster in a Zinc paddle-wheel (e.g., in Zn-HKUST-1, a single node (e.g., in ZIFs), Zn₄O (e.g., in IRMOFs), or to a totally new configuration.

The diversity in metal chemistry was further reduced by the choice of researchers and/or limitations in MOF structure assembly algorithms. For example, some of the hypothetical MOF databases are deliberately focused on specific sub-classes of MOFs to systematically investigate structure-property relationships. For example, the study by Gomez Gualdrón et al. that focuses on generating stable MOFs using Zirconium based metal nodes, Witman et al. on 1-D rod MOFs featuring open-metal sites, and Moosavi et al. on ZIFs with various functional groups and underlying nets. Lastly, in silico assembly of MOFs possessing complex nodes that are connected via multiple linkers, especially on a low-symmetry net, is still challenging for the current structure generation algorithms”

3) How common are the nodes presented in the SI? That is how many CoRE MOFs have them? What kind of linkers are associated with them?

Authors reply:

The information of the presented nodes is added to the revised SI.

3) About the discussion around Figure 4. Specifically, the ARABG database. In the cited publication, Anderson et al report variable importance for some CO₂ adsorption metrics (selectivity, working capacity), with a pore descriptor having the largest importance (kind of in line with the variable importance here for BW-20K). These are not the CO₂@0.15bar adsorption metrics the authors are using here, but it does make me wonder how sensitive general conclusions regarding whether chemistry or pore geometry are more important are to how the chemical and geometrical descriptors are defined, as well as the type of machine learning method.

Authors reply:

The reviewer points to a very interesting factor that is the difference in descriptors. Our perspective in developing descriptors has been an experimental point of view; follow the intuition of the chemists. However, some of these descriptors can be related, for example, adding a functional group can change the shape of pores. So, then it is perfectly possible that for one set of properties this is seen as a pore shape effect and in the other a change of the functional group. Of course, both results are consistent within the framework of the particular descriptors, but we agree that the readers should be aware of this. We therefore added:

“In developing these descriptors, it is impossible to completely separate the different effects and scopes. For example, for some MOFs adding a functional group can completely change the pore shape. Hence, depending on the details of the different types of descriptors this may be seen as mainly pore-shape effect, while other sets of descriptions will assign it as functional-group effect.”

4) One implication of the discussion in this work is that simply trying to get high number of structures in a database for the sake of getting a high number is not the best use of resources. A more targeted MOF generation for the application of choice may be the smarter strategy. For example, the ToBaCCo database seems to have been originally explored for H₂, CH₄ and Xe/Kr, which are not expected to be affected by functionalization, making it OK for the database not to have functional groups. On the other hand, the ARABG database seems small but emphasizes on functional groups, which base on the discussion of this paper seems very important for CO₂ adsorption. For ARABG, machine learning was done with a small set (considering typical databases), so my question is whether in the current work the authors have unveiled some sort of, at least semi-quantitative, insights or rules of thumb about critical database size (assuming maximum diversity) beyond which machine learning models will not learn much more. (This perhaps relates to the kind of work presented in Fig. 5)

Authors reply:

The learning curves of machine learning models, in particular those from k-nearest-neighbors (kNN) models, can be a good indication of the optimal coverage of the design space and minimum database size. In these curves, the accuracy of machine learning is plotted with respect to the number of training points. The minimum number of structures would be the point which the accuracy of the model does not improve further. Of course, this number is absolutely application dependent; for example, for an application like CO₂ adsorption at high pressures, the single most important variable is pore volume, and therefore, a small set of materials with diverse pore volume would be sufficient to get a good coverage of the design space. Therefore, one can develop such set for the application of interest using the methodology and tools introduced in this study.

In the revised version, we included multiple learning curves for different application in SI, and also we refer the readers to this discussion in the SI:

“Besides, the diversity in training set lead to a more efficient learning. In supplementary materials, we show the learning curves that demonstrate the models trained on the diverse set have systematically lower error than the ones trained using biased databases. The number of training points in which the learning curves plateau can be an indication of the minimum number of structures for optimal coverage of the design space for a particular application. This number is obviously proportional to the complexity of the material property, i.e., how many materials characteristics are affecting the materials properties.”

4) I think it would be a good idea to present histograms in the supplementary information as complementary way to present the information given in Figure 2 and Figure 3. Some readers may find those histograms easier to interpret.

Authors reply:

As the reviewer requested, we performed a full exploratory data analysis that includes histograms, various type of correlation analysis, data ranges, data types, etc. Since the number of features and databases are large and would occupy too large space in SI, we only included these analyses to the materials cloud archive.

5) Arguably, one way to read Fig. 4 could be that the screening the CoRE MOFs fails to capture that linker functionalization is important for CO₂ adsorption as noted by screening of ARABG or BW-20K, or as thought of my experimentalists who, for example, consider strategies such as amino functionalization to boost CO₂ sorption. In inspecting Fig S13 and S14, I can't help but notice that the CoRE MOFs fail to cover an area of functional groups that is covered by the BW database (I don't know about ARABG). Moreover, the area the CoRE MOFs do cover tend toward the area that ToBaCCo covers, which we know does not really have functional groups. So I wonder if the CoRE MOFs are just lacking in those key functional groups that truly have an impact on CO₂ adsorption, which allow metal chemistry to take such an outsized role.

Authors reply:

Yes, the CoRE MOFs do not exploit the functional groups as much as the other databases, e.g, ARABG-DB or BW-DB, do. Since these are relative importance of variables, one can say the metal chemistry is much better explored in CoRE in comparison to the functional groups. The reason we find some sort of functional groups in ToBaCCo is that we assign any heterogenous atom on linker that is not carbon or hydrogen and does not have a bond to the metal to be a functional group. Such assignment for ToBaCCo database will label those nitrogen atoms on the linker that are substituting carbons. Therefore, the reviewer's guess is right that there are many similarities in functional groups of ToBaCCo and CoRE MOFs.

In the revised version, we add this sentence to emphasis on this:

“Also, since these values are the relative importance, one can argue that in CoRE-2019 MOFs, the functional groups were not exploited as much as metal chemistry.”

6) The authors state: “This explains the relatively poor performance in prediction of CO₂ adsorption properties using only geometric descriptors as electrostatic interaction plays a crucial role.” Based on an earlier sentence, I think the authors may have meant to say: “(...) prediction of CO₂ adsorption properties at low pressure”.

Authors reply:

Corrected in the revised manuscript.

Response to Reviewer #3

The authors have created a well written paper that describes a new way to classify and organize the many MOFs available in popular databases, and to study their diversity in doing so. In addition to being well written, the paper is well organized and the results are clearly presented. I think this paper can be accepted essentially without revisions.

That being said, notions of MOF diversity are necessarily (?) subjective. It's hard to imagine how one can be write or wrong when measuring MOF diversity, so the aim is more to be more or less useful, but that will then depend on what applications MOF scientists are interested in. To their credit, the authors do get into applications and comment on the utility of their measures for those purposes... but it would have been nice to see an at least short discussion in the introduction that acknowledges this fundamental subjectivity. Conversely, if I am wrong and MOF diversity can be objectively measured, then I would certainly want the authors to explain that early in the paper!

In summary, unsurprisingly good work from great authors. Looking forward to seeing it published!

Authors reply:

The reviewer is right that diversity is only objective with respect to an application. Our methodology, however, can objectively measure diversity for a given application. However, we do realize that in Figure 6, we “subjectively” assumed that pore geometry is important, but then we do argue that if pore geometry is important figure 6 gives objectively that structures that were very different in pore geometry compared to the know structures at that time. To clarify this, we added the underlined text:

“An interesting historical perspective is shown in Figure 6., in which we plot as metric of novelty of the discovered materials the distance to the geometry descriptor of the previously discovered materials. Here, we assume pore geometry is the important factor of interest.”

In addition, we added to the conclusion:

“In this work we aim to address the question whether a new material adds novelty. We try to develop transparent and objective criteria to quantify how different a novel material is with respect to the state of the art. However, as soon as we use this for a particular application, it becomes subjective. For example, if Figure 6 we selected novelty of pore geometry. This measure by definition completely ignores, for example, the importance of making the first MOF with a particular metal, which might be the single most important factor for, say, an application related to catalysis.”